# Test Case Selection through Novel Methodologies for Software Application Developments

Sekar Kidambi Raju [1], Sathiamoorthy Gopalan [2], S. K. Towfek [3,4,*], Arunkumar Sukumar [1], Doaa Sami Khafaga [5], Hend K. Alkahtani [6,*] and Tahani Jaser Alahmadi [6]

[1] School of Computing, SASTRA Deemed University, Thanjavur 613401, India; sekar_kr@cse.sastra.ac.in (S.K.R.); arunkumar@cse.sastra.ac.in (A.S.)
[2] Department of Maths, SASHE, SASTRA Deemed University, Thanjavur 613401, India; sami@maths.sastra.ac.in
[3] Computer Science and Intelligent Systems Research Center, Blacksburg, VA 24060, USA
[4] Department of Communications and Electronics, Delta Higher Institute of Engineering and Technology, Mansoura 35111, Egypt
[5] Computer Sciences, College of Computer and Information Sciences, Princess Nourah Bint Abdulrahman University, P.O. Box 84428, Riyadh 11671, Saudi Arabia; dskhafaga@pnu.edu.sa
[6] Department of Information Systems, College of Computer and Information Sciences, Princess Nourah Bint Abdulrahman University, P.O. Box 84428, Riyadh 11671, Saudi Arabia; tjalahmadi@pnu.edu.sa
* Correspondence: sktowfek@jcsis.org (S.K.T.); hkalqahtani@pnu.edu.sa (H.K.A.)

**Abstract:** Test case selection is to minimize the time and effort spent on software testing in real-time practice. During software testing, software firms need techniques to finish the testing in a stipulated time while uncompromising on quality. The motto is to select a subset of test cases rather than take up all available test cases to uncover most bugs. Our proposed model in the research study effort is termed SCARF-RT, which stands for Similarity coefficient (SC), Creating Acronyms, Regression test (RT), and Fuzzy set (FS) with Dataset (DS). Clustering of test cases using ranking and also based on similarity coefficients is to be implemented. This research considered eleven different features for clustering the test cases. Two techniques have been used. Firstly, each cluster will, to a certain extent, encompass a collection of distinct traits. Depending on the coverage of the feature, a cluster of test cases might be chosen. The ranking approach was used to create these groupings. The second methodology finds similarity among test cases based on eleven features. Then, the maxmin composition is used to find fuzzy equivalences upon which clusters are formed. Most similar test cases are clustered. Test cases of every cluster are selected as a test suite. The outcomes of this research show that the selected test cases based on the proposed approaches are better than existing methodologies in selecting test cases with less duration and at the same time not compromising on quality. Both fuzzy rank-based clustering and similarity coefficient-based clustering test case selection approaches have been developed and implemented. With the help of these methods, testers may quickly choose test cases based on the suggested characteristics and complete regression testing more quickly.

**Keywords:** test case selection; cluster; fuzzy rank coefficients; regression test; similarity coefficient; fuzzy equivalences

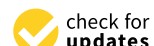



## 1. Introduction

In software engineering, testing is performed just before deployment. Many sets of inputs are created for testing, and software is tested with these inputs to see that the output is as desired. If not, the bugs are reported to the developer to correct the errors mentioned. But, while correcting errors, a new set of errors may be introduced. So, the testing needs to be conducted again. This approach of retesting is called a regression test. Re-exercising all test cases is impossible during regression tests, as it is a highly time-consuming task. In test case scenarios, white-box testing provides a clear version of test cases. The research

developed a white-box testing approach that prioritizes, selects, and minimizes in the context of reusability under constraints in the scope of usability. Their approach is a boost to existing approaches [1]. Instead, some of the test cases alone need to be selected, which will plausibly catch up with errors. The paper develops a test case selection methodology as a constrained search-based optimization task using requirements coverage as a fitness function to be maximized. They used binary-constrained particle swarm optimization as a base technique [2]. The research work proposed an approach to select test cases in database applications. They designed classification tree models using a similarity base towards black-box testing. Also, a new set of test cases for modified code sections is to be included in the regression test. Moreover, the team lead subjectively decides the number of iterations in regression tests.

So, selecting test cases to catch significant errors with minimal time duration has become necessary for the software industries. And the major challenge in this approach is retaining the software's quality. The research work proposed an approach to select test cases in database applications. They designed classification tree models using a similarity base towards black-box testing [3]. The paper proposed a test case allocation method using fuzzy inference [4]. The research article proposed a weighted attribute-based strategy by conducting more than one clustering iteration using weighted execution profiles [5]. Therefore, uncovering more errors, especially the critical and showstopper errors with less time duration, is difficult. And this is the research topic of the current era. Researchers have devised techniques based on code. Such techniques are delayed and known to be the white-box approach to software testing. It considers the code flow during execution. The research proposed regression test selection using program dependence graphs for service-oriented workflow applications [6]. The paper developed two selective regression testing methods to evaluate the effectiveness of the Rosenblum-Weyuker (RW) model for predicting cost-effectiveness [7]. Also, the data flow across the different paths of code is considered. Such a transparent view of the coding gives clear visibility toward finding bugs and, hence the name. The literature depicts clearly that many researchers rely on design notations and models upon which test cases were generated. The formation of test cases is being achieved for finding bugs, whose backbone is sheer design models. Mirarab et al. [8] proposed an approach forming an integer linear programming problem using two different coverage-based criteria to select a subset of test cases and then order them using a greedy algorithm to maximize the coverage. Fujiwara et al. [9]. This approach neither relies on code nor depends on querulents. Hence, it sits amidst white-box and black-box approaches and is termed the grey-box approach in software tests. Yet another approach adopted by practitioners and researchers is generating test cases concerning functional and nonfunctional requirements. The code is completely blacked out as part of testing and hence named a black-box testing approach. Our approach is based on features of the software test in selecting test cases. These features are not picked from implicit or explicit requirements. Instead, these features are marked exclusively by software testers. So, the proposed test case selection methodology can be applied to any test approach such as black box, white box, or grey box. The skilled testers across various software building industries set the order of importance of these features.

Our contributions in this work:

- To introduce an inventive hybrid grey-box testing methodology that effectively combines white-box and black-box strategies.
- Develop a distinctive feature-based test case selection approach, emphasizing vital aspects of the software designated by proficient testers without reliance on explicit or implicit requirements.
- Develop pioneering quantum-based test case selection techniques, employing feature coverage concepts and rank-based coefficients to optimize the relationship between features and test cases.

The innovative hybrid grey-box testing methodology seamlessly integrates the strengths of both white-box and black-box testing strategies, optimizing the testing process for en-

hanced efficiency and thoroughness. The black-box testing reflects real-world usage scenarios and user perspectives, ensuring a comprehensive assessment of system behavior. By combining these approaches, our methodology leverages the advantages of white-box testing's precision and black-box testing's comprehensiveness, offering a holistic evaluation of the software's functionality, security, and reliability. The innovative hybrid grey-box testing sustains the model's stability, harmonizing white-box understanding and black-box versatility, ensuring a resilient and dependable performance across diverse conditions.

The distinctive feature-based approach prioritizes critical aspects of the software, identified and designated by proficient testers, bypassing the need for explicit or implicit requirements. Leveraging their expertise and insights, testers meticulously pinpoint significant features integral to the software's functionality and user experience. This methodology advocates for a proactive testing strategy, ensuring that crucial functionalities and user interactions are thoroughly assessed, even without exhaustive requirements documentation. The distinctive feature-based test case selection approach, guided by proficient testers, fortifies the model's stability without relying on explicit or implicit requirements, enhancing software robustness and reliability.

Innovate test case selection in quantum computing through groundbreaking methods using feature coverage principles and rank-based coefficients. By intricately optimizing the interplay between features and test cases, we aim to significantly enhance testing efficiency and accuracy and ultimately pave the way for more effective and reliable quantum computing systems, marking a significant leap in the realm of quantum technology. The pioneering quantum-based test case selection leverages feature coverage and rank-based coefficients, optimizing the feature-test case relationship, bolstering the model's stability, and enhancing software reliability.

The stated order of importance over the features in our proposed work is the renowned, profound, and benchmarked order, which every tester practices while optimizing test cases. The research work-related features and test cases through rank-based coefficients. In the existing market, many other techniques prevail. In this paper, we have developed methods for test case selection based on the quantum of the feature coverage. Altogether, these proposed techniques help to improve software testing effectiveness. This selection process yields an optimal set of test cases, especially in regression tests.

*The Scientific Novelty of the Research Work*

The effective selection of test cases is crucial in software testing since it helps to reduce the time and effort needed for full testing while maintaining high standards. Software companies look for methods to speed up the testing process without sacrificing quality to meet this demand. This research focuses on implementing test case clustering utilizing ranking and similarity coefficients to find the most faults in a shorter time. The experimentation involves considering eleven distinct features to cluster the test cases, employing two methodologies. In the first methodology, clusters are formed based on covering specific features up to a certain percentage. This ranking methodology aids in selecting the test case clusters. The second methodology determines test case similarity using the eleven features, and fuzzy equivalences are derived through maxmin composition.

Novelty 1: This research pioneers an innovative approach to test case clustering using a unique combination of fuzzy rank-based clustering and similarity coefficients, allowing for efficient selection of test cases based on specific features and significantly accelerating regression testing without compromising quality.

Novelty 2: The experimentation in this study demonstrates the superiority of the proposed methodologies over existing approaches by utilizing eleven distinct features to cluster test cases, achieving higher fault detection rates in a shorter duration. This advancement offers a practical solution for software companies seeking enhanced testing efficiency.

For your convenience, the dataset and coding information are accessible in the Supplementary Materials.

The coming sections discuss related work, motivational examples, empirical evaluation and metrics, results and discussions, future work, and conclusions.

## 2. Literature Review

Introduced the "partial W" method, which provides a logical link between several finite state machine model test methods, providing general applicability in selecting test cases. Mansour et al. [10] have compared five regression test selection algorithms: simulated annealing, reduction, slicing, data flow, and firewall algorithms. The criteria they used are the number of selected test cases, execution time, precision, inclusiveness, preprocessing requirements, maintenance, level of testing, and type of approach. Mansour et al. [11] proposed three test selection methods. The first method is based on modification and its effects. The second method omits tests that do not cover modification. The third method reduces the number of test cases selected by omitting non-modification revealing tests from the initial suite. Lee et al. [12] proposed a set covering problems using an enhanced zero-one optimal path set selection method to select test cases in structural programming. Graves et al. [13] experimented to examine the relative costs and benefits of several regression test selection techniques. Rothermel et al. [14] presented a framework for evaluating regression test selection techniques regarding inclusiveness, precision, efficiency, and generality. Erikrogstad et al. [15] explored the cost and effectiveness of various approaches and their combination for regression testing of database applications using classification tree models of the input domain.

### 2.1. Various Test Cases and Methodologies

Zheng et al. [16] developed a multi-objective evolutionary algorithm, evaluating classic greedy and non-dominated sorting genetic algorithms II. This technique provides a range of solutions with trade-offs between cost and coverage. Rafaqutkazmi et al. [17] scrutinized 47 regression test case selection articles comprising seven cost measures, 13 coverage types, and five fault detection metrics. Gr Rothermel et al. [18] developed an algorithm to select test cases based on the modified versions of the program using a control flow graph. Francisco et al. [19] have devised a similarity technique to select test cases for the modified versions of the program based on state diagrams. Briand et al. [20] scrutinized UML designs and devised a technique for test case selection based on their changes. Rapps et al. [21] deployed input selection techniques based on the data flow chain across the program paths. Harrold et al. [22] implemented a methodology for selecting test cases based on requirements coverage cardinality, reducing the test suite size. Raju et al. [23] analyzed metrics in testing and chose test cases accordingly for regression tests. Wong et al. [24] developed a hybrid technique for selecting test cases based on code coverage, fault classification, execution profiles, and program modifications. L Yu et al. [25] devised classification models to classify test cases based on defect types they captured and dynamic programming to get optimal selection solutions. In a distributed cloud environment, employing multi-objective criteria is essential for efficient test case selection and prioritization. Balancing objectives like performance, resource utilization, and fault tolerance ensures comprehensive testing for optimal system reliability and functionality [26]. This study focuses on the perceived impact of an offshore aquaculture area in southeastern Portugal, the Armona Pilot Production Aquaculture Area (APPAA). The infrastructure creation aimed to stimulate local employment opportunities in seafood production and improve finfish and shellfish production resilience [27]. This research paper proposes an expert system based on a fuzzy logic model to analyze the dynamics of sustainable livestock production systems. The objective is to address the complexities and uncertainties inherent in livestock production while aiming for sustainable practices [28,29]. The literature survey in the test case selection domain is summarized in Table 1.

**Table 1.** Literature survey in the domain of test case selection.

| S.No | Year | Title of Paper | Methodology | Strength | Weakness |
|---|---|---|---|---|---|
| 1 | 1985 | Selecting software test data using data flow information | Definition—Use graph | Detects Dependencies | Complexity |
| 2 | 1991 | Test Selection Based on Finite State Models | Finite state models | Automation | Interpretability Challenges |
| 3 | 1993 | A methodology for controlling the size of a test suite | Association between requirement and test case | Efficiency in Test Execution | Potential Coverage Gaps |
| 4 | 1996 | Analyzing regression test selection techniques | Linear equation, path analysis, data flow, dependency graph, modification, firewall, cluster techniques | Enhanced Test Suite Efficiency | Overhead in Selection Process |
| 5 | 1997 | A safe efficient Regression Test selection technique | Control flow graph | Reduced Test Execution Time | Initial Setup Complexity |
| 6 | 1997 | A Study of Effective Regression Testing in Practice | Modification-based test selection minimized or prioritized test sets | Size reduction, precision, and recall are analyzed | Initial Setup Complexity |
| 7 | 2000 | An optimal representative set selection method | Optimal representative set and optimal path set selection | Targeted Scenario Coverage | Possible Missed Dependencies Detection |
| 8 | 2001 | An empirical study of regression test selection techniques | Comparing all test selection methods | Data-Driven Testing Approach | Potential Generalization Limitations |
| 9 | 2001 | Comparison of regression test selection algorithms | Comparison based on eight different criteria | Comprehensive Algorithmic Performance Evaluation regression testing | Sensitivity to Test Cases |
| 10 | 2001 | Empirical studies of a Prediction model for Regression Test Selection | A prediction model based on coverage | Predictive Capability for Selection | Model Training Overhead and Complexity |
| 11 | 2002 | Reduction-based methods and metrics for selective regression testing | Modification and precisely based reduction | Optimized Testing Effort Allocation | Reliance on Reduction Techniques |
| 12 | 2009 | Automating regression test selection based on UML designs | Mapping between UML design changes and classification of test cases | Efficient Utilization of UML | Initial Setup and Integration Challenges |
| 13 | 2010 | Time-constrained test selection for regression testing | Data mining to select test cases and dynamic programming to find optimal test cases | Timely and Prioritized Test Selection | Sensitivity to Time Constraints |
| 14 | 2012 | Size-constrained regression test case selection using multicriteria optimization | Selection using integer linear programming problem and prioritization using greedy algorithm | Effective Utilization of Criteria | Selection Criteria Sensitivity |
| 15 | 2013 | Search-based constrained test case selection using execution effort, expert systems with applications | Constrained search-based optimization | Utilizes Execution Effort Criteria | Potential Selection Bias Impact |
| 16 | 2013 | Test case selection for black-box regression testing of database applications, Information, and Software Technology | Similarity-based test selection algorithm | Targeted Black-Box Test Selection | Limited Coverage of Scenarios |
| 17 | 2014 | A weighted attribute-based strategy for cluster test selection | Weighted attributes strategy | Utilizes Weighted Attribute Considerations | Sensitivity to Attribute Weighting |

**Table 1.** *Cont.*

| S.No | Year | Title of Paper | Methodology | Strength | Weakness |
|------|------|----------------|-------------|----------|----------|
| 18 | 2014 | System regression test planning with a fuzzy expert system | fuzzy expert system based on features | Comprehensive Regression Test Planning | Potential Model Complexity Impact |
| 19 | 2014 | Measurement and Analysis of Test Suite Volume Metrics for regression Testing | Regression test on various applications | Quantitative Analysis of Test Volume. | Data Sensitivity to Variability |
| 20 | 2015 | A Novel Method for Allocating Software Test Cases | Fuzzy inference on software operational profile | Innovative Test Case Allocation Method | Sensitivity to Allocation Criteria |
| 21 | 2016 | Scope-aided test prioritization, selection, and minimization for software reuse | Code coverage testing | Efficient Test Suite Minimization | Potential Overlooking of Important Scenarios |
| 22 | 2016 | Cost-effective strategies for the regression testing of database applications | Classification tree model | Focus on Database Application Specifics | Sensitivity to Cost Factors |
| 23 | 2016 | Multi-objective optimization for regression testing | Multi-objective evolutionary algorithm | Balanced Test Objective Fulfillment coverage | Trade-offs Among Multiple Objectives |
| 24 | 2016 | Full modification coverage through automatic | similarity analysis | Automated Modification Tracking and Testing | Potential Overlooked Edge Cases |
| 25 | 2017 | Effective Regression Test Case Selection | 47 empirical studies, seven cost measures,13 coverage types, five fault detection metrics | Targeted Test Case Coverage. | Potential Missed Regression Scenarios |
| 26 | 2017 | Optimal control-based regression test selection for service-oriented workflow applications | Optimal control based on dependence graph | Efficient Test Case Selection | Sensitivity to Workflow Changes |

## 2.2. Analysis of a Literature Work to Find the Strengths and Weaknesses in Research

This study's key findings encompassed a thorough analysis of software testing methodologies, focusing on optimizing efficiency and effectiveness. The research emphasized detecting dependencies and implementing automation, which significantly enhanced test execution efficiency and reduced overall test execution time. Additionally, this study showcased the importance of a data-driven testing approach and a comprehensive algorithmic performance evaluation for regression testing. The predictive capability for test case selection and the efficient allocation of testing efforts were critical factors in achieving targeted scenario coverage and prioritized test selection. The utilization of UML, innovative test case allocation methods, and automated modification tracking were noted to contribute to an efficient testing process. Common strengths observed were the comprehensive regression test planning, balanced test objective fulfillment coverage, and a focus on database application specifics, ensuring a robust and optimized testing effort allocation. The integration of a quantitative analysis of test volume and efficient test suite minimization further highlighted the research's commitment to precision and efficiency in testing practices.

The limitations encompass challenges in accurately detecting complex and dynamically changing dependencies, potential resource constraints affecting concurrent test executions for efficient test execution, and the complexity in achieving optimal efficiency for heterogeneous test suites, leading to enhanced test suite efficiency limitations. Reducing test execution time faces constraints due to application intricacies and the need to main-

tain comprehensive testing coverage, and analyzing size reduction, precision, and recall presents difficulty in finding the perfect balance between these aspects and handling diverse data sources for data-driven testing. Targeted scenario coverage encounters difficulty in identifying and covering all relevant and possible scenarios, while comprehensive algorithmic performance evaluation in regression testing may struggle to predict all impacted areas accurately. Predictive capability for selection and optimized testing effort allocation face uncertainties and potential inaccuracies in prediction and difficulty in accurately allocating efforts amidst evolving project priorities, respectively. Integration complexities challenge the effective utilization of UML, and achieving timely and prioritized test selection can be subjective and prone to biases. The effective utilization of criteria encounters challenges in adapting to changing requirements and potential conflicts in criteria. Quantitative analysis of test volume may face variability in effectiveness, and innovative test case allocation methods struggle with implementation complexities and justifying effectiveness. Efficient test suite minimization faces constraints in achieving significant efficiency gains for already optimized suites, and focusing on database application specifics may be limited by the difficulty in adapting the focus to diverse applications. Balancing test objective fulfillment coverage encounters subjectivity and challenges in satisfying all objectives simultaneously. Automated modification tracking and testing may be complex to fully automate tracking for rapidly changing software. Targeted test case coverage and efficient test case selection face challenges in achieving complete coverage and balancing efficiency with thoroughness, respectively.

## 3. Motivational Example

For the research work illustration, we have taken a training set which is shown in Table 2. The training set available in Table 2 contains 15 different real-time test cases. These test cases are a subset of the test suites for the online banking software system. Eleven different features were considered for this research work, as shown in Table 2. The 11 different features are ordered in the training set in such a way that they signify the order of importance concerning testing. This order of importance among features is a gathered knowledge from the test experts of the industries, which they follow in Table 2. The research work uses the proposed model SCARF-RT to identify the best cluster for software test case selection.

**Table 2.** Sample data set for test cases.

| TC | C1 | C2 | C3 | C4 | C5 | C6 | C7 | C8 | C9 | C10 | C11 |
|------|----|----|----|----|----|----|----|----|----|-----|-----|
| TC1  | VH | H  | L  | M  | VL | VL | VH | VH | M  | H   | M   |
| TC2  | H  | M  | L  | VL | L  | M  | VH | H  | M  | VL  | L   |
| TC3  | VH | H  | M  | L  | VL | M  | VH | VH | M  | VL  | M   |
| TC4  | M  | M  | M  | M  | L  | L  | H  | M  | M  | M   | M   |
| TC5  | VH | H  | H  | M  | M  | H  | L  | VL | M  | H   | H   |
| TC6  | H  | VH | M  | M  | L  | VL | L  | M  | M  | VH  | H   |
| TC7  | VH | H  | M  | M  | M  | L  | VL | L  | H  | H   | M   |
| TC8  | VL | L  | H  | L  | M  | L  | L  | VL | M  | H   | M   |
| TC9  | VH | H  | M  | M  | M  | L  | VH | H  | H  | H   | M   |
| TC10 | M  | M  | M  | M  | M  | L  | VL | L  | M  | M   | M   |
| TC11 | VH | H  | M  | M  | M  | L  | H  | H  | H  | H   | M   |
| TC12 | VH | H  | H  | H  | H  | L  | VL | L  | H  | H   | M   |
| TC13 | VH | H  | M  | H  | M  | L  | L  | L  | H  | H   | M   |
| TC14 | M  | M  | M  | M  | M  | L  | VL | L  | H  | L   | M   |
| TC15 | VH | VH | VH | H  | H  | L  | L  | L  | H  | H   | M   |

Here, in Table 2, twelve test cases are considered, and eleven features are taken. If we want to test the test cases of the software, we have to consider the features of software applications. Here, the features are nonfunctional activities like Critical bugs, Requirements covered, Time, KLOC covered, Bugs detected, Length critical requirements

covered, Customer priority, Fault proneness, Requirements volatility, and Implementation complexity. In the fuzzy, two important jargon are valued: one is linguistics, and another is intuitional-tics; mainly in linguistics, we measure everything as very low (VL), low (L), medium (M), high (H), and very high (VH), all these fuzzy terms are converted to intuitional-tics values from 1 to 5. In our scenario, very low-1, low-2, medium-3, high-4, and very high-5. In Table 2, the test cases are available from 1 to 15. Each test case has features from C1 to C11 and is given linguistic scores like very low-1, low-2, and medium-3 for the corresponding symbols. Each test case has 11 parts to be tested. This is how Table 2 was formulated.

In fuzzy, linguistic and intuitive values are offered. The linguistic words are very low (VL), low (L), medium (M), high (H), and very high (VH). Linguistic values are offered as very low-1, low-2, medium-3, high-4, and very high-5 to analyze words in decision support systems. Depending on the nature of the application, the values can be adjusted to a 5 or 10 scale.

Legends: C1—Critical bugs: This metric refers to the number of critical defects or issues discovered during testing. C2—Requirements covered: This metric measures how much the test cases cover the specified requirements. C3—Time: This metric evaluates the time to execute the test cases and measures the testing effort. C4—LOC (Lines of Code) covered: This metric determines the number of lines of code exercised or covered by the test cases. C5—Bugs detected: This metric tracks the total number of defects discovered during testing, including all severity levels. C6—Average word length: It evaluates the number of test steps, input data variations, or conditions covered by each test case. C7—Critical requirements covered: Similar to C2, this metric specifically focuses on critical needs and assesses the coverage of test cases related to those requirements. C8—Customer priority: This metric prioritizes test cases based on customer requirements and preferences. C9—Fault proneness: This metric aims to predict the likelihood of defects occurring in specific areas of the software based on past defect data. C10—Requirements volatility: Requirements volatility measures the frequency of changes to the software's requirements during the testing process. C11—Implementation complexity: This metric assesses the complexity of the software implementation and its impact on testing efforts.

I considered the 11 features of my empirical study in my research study. These are critical aspects of my application. The various features for various applications may be considered and included. These 11 features produce excellent results with high precision, as seen in the tables.

In Table 3, for every linguistic value, the corresponding intuitional-tics values of very low (VL)-1 and low (L)-2 like this, the scale values from 1 to 5 are substituted, respectively.

**Table 3.** Intuitional-tics values.

| TC | C1 | C2 | C3 | C4 | C5 | C6 | C7 | C8 | C9 | C10 | C11 |
|----|----|----|----|----|----|----|----|----|----|-----|-----|
| TC1 | 5 | 4 | 2 | 3 | 1 | 1 | 5 | 5 | 3 | 4 | 3 |
| TC2 | 4 | 3 | 2 | 1 | 2 | 3 | 5 | 4 | 3 | 1 | 2 |
| TC3 | 5 | 4 | 3 | 2 | 1 | 3 | 5 | 5 | 3 | 1 | 3 |
| TC4 | 3 | 3 | 3 | 3 | 1 | 1 | 4 | 3 | 3 | 3 | 3 |
| TC5 | 5 | 4 | 4 | 3 | 3 | 4 | 2 | 1 | 3 | 4 | 4 |
| TC6 | 4 | 5 | 3 | 3 | 2 | 1 | 2 | 3 | 3 | 5 | 4 |
| TC7 | 5 | 4 | 3 | 3 | 3 | 2 | 1 | 2 | 4 | 4 | 3 |
| TC8 | 1 | 2 | 4 | 2 | 3 | 2 | 2 | 1 | 3 | 4 | 3 |
| TC9 | 5 | 4 | 3 | 3 | 3 | 2 | 5 | 4 | 4 | 4 | 3 |
| TC10 | 3 | 3 | 3 | 3 | 3 | 2 | 1 | 2 | 3 | 3 | 3 |
| TC11 | 5 | 4 | 3 | 3 | 3 | 2 | 4 | 4 | 4 | 4 | 3 |
| TC12 | 5 | 4 | 4 | 4 | 4 | 2 | 1 | 2 | 4 | 4 | 3 |
| TC13 | 5 | 4 | 3 | 4 | 3 | 2 | 2 | 2 | 4 | 4 | 3 |
| TC14 | 3 | 3 | 3 | 3 | 3 | 2 | 1 | 2 | 4 | 2 | 3 |
| TC15 | 5 | 5 | 5 | 4 | 4 | 2 | 2 | 2 | 4 | 4 | 3 |

In Table 4, the values zero and one are assigned with a threshold. If the numbers are 3 or above, we can supply 1; otherwise, we can provide 0. Table 4 was created using the variables from Table 3.

**Table 4.** Fuzzy equivalence values.

| TC | C1 | C2 | C3 | C4 | C5 | C6 | C7 | C8 | C9 | C10 | C11 |
|----|----|----|----|----|----|----|----|----|----|-----|-----|
| TC1 | 1 | 1 | 0 | 1 | 0 | 0 | 1 | 1 | 1 | 1 | 1 |
| TC4 | 1 | 1 | 1 | 1 | 0 | 0 | 1 | 1 | 1 | 1 | 1 |
| TC5 | 1 | 1 | 1 | 1 | 1 | 1 | 0 | 0 | 1 | 1 | 1 |
| TC6 | 1 | 1 | 1 | 1 | 0 | 0 | 0 | 1 | 1 | 1 | 1 |
| TC7 | 1 | 1 | 1 | 1 | 1 | 0 | 0 | 0 | 1 | 1 | 1 |
| TC8 | 0 | 0 | 1 | 0 | 1 | 1 | 1 | 0 | 1 | 1 | 1 |
| TC9 | 1 | 1 | 1 | 1 | 1 | 0 | 1 | 1 | 1 | 1 | 1 |
| TC10 | 1 | 1 | 1 | 1 | 1 | 0 | 0 | 1 | 1 | 1 | 1 |
| TC12 | 1 | 1 | 1 | 1 | 1 | 0 | 0 | 0 | 1 | 1 | 1 |
| TC13 | 1 | 1 | 1 | 1 | 1 | 1 | 1 | 1 | 0 | 0 | 0 |
| TC14 | 1 | 1 | 1 | 0 | 1 | 1 | 1 | 1 | 0 | 0 | 0 |
| TC15 | 1 | 1 | 1 | 1 | 1 | 1 | 1 | 1 | 0 | 0 | 0 |

Table 5, row-wise total be taken for all the rows from TC1 to TC15. In the rows, 1's are added as such. In Table 6, the row-wise total will be arranged in the order called ranking. Through Table 6, it is possible to understand the ideology.

**Table 5.** Row-wise total.

| TC | C1 | C2 | C3 | C4 | C5 | C6 | C7 | C8 | C9 | C10 | C11 | Total |
|----|----|----|----|----|----|----|----|----|----|-----|-----|-------|
| TC1 | 1 | 1 | 0 | 1 | 0 | 0 | 1 | 1 | 1 | 1 | 1 | 8 |
| TC2 | 1 | 1 | 0 | 0 | 0 | 1 | 1 | 1 | 1 | 0 | 0 | 6 |
| TC3 | 1 | 1 | 1 | 0 | 0 | 1 | 1 | 1 | 1 | 0 | 1 | 8 |
| TC4 | 1 | 1 | 1 | 1 | 0 | 0 | 1 | 1 | 1 | 1 | 1 | 9 |
| TC5 | 1 | 1 | 1 | 1 | 1 | 1 | 0 | 0 | 1 | 1 | 1 | 9 |
| TC6 | 1 | 1 | 1 | 1 | 0 | 0 | 0 | 1 | 1 | 1 | 1 | 8 |
| TC7 | 1 | 1 | 1 | 1 | 1 | 0 | 0 | 0 | 1 | 1 | 1 | 8 |
| TC8 | 0 | 0 | 1 | 0 | 1 | 1 | 1 | 0 | 1 | 1 | 1 | 7 |
| TC9 | 1 | 1 | 1 | 1 | 1 | 0 | 1 | 1 | 1 | 1 | 1 | 10 |
| TC10 | 1 | 1 | 1 | 1 | 1 | 0 | 0 | 1 | 1 | 1 | 1 | 9 |
| TC11 | 1 | 1 | 1 | 1 | 1 | 0 | 1 | 1 | 1 | 1 | 1 | 10 |
| TC12 | 1 | 1 | 1 | 1 | 1 | 0 | 0 | 0 | 1 | 1 | 1 | 8 |
| TC13 | 1 | 1 | 1 | 1 | 1 | 1 | 1 | 1 | 0 | 0 | 0 | 8 |
| TC14 | 1 | 1 | 1 | 0 | 1 | 1 | 1 | 1 | 0 | 0 | 0 | 7 |
| TC15 | 1 | 1 | 1 | 1 | 1 | 1 | 1 | 1 | 0 | 0 | 0 | 8 |

**Table 6.** Ranking made according to the row-wise total in a descending order.

| TC | C1 | C2 | C3 | C4 | C5 | C6 | C7 | C8 | C9 | C10 | C11 | Row Total |
|----|----|----|----|----|----|----|----|----|----|-----|-----|-----------|
| TC9 | 1 | 1 | 1 | 1 | 1 | 0 | 1 | 1 | 1 | 1 | 1 | 10 |
| TC11 | 1 | 1 | 1 | 1 | 1 | 0 | 1 | 1 | 1 | 1 | 1 | 10 |
| TC4 | 1 | 1 | 1 | 1 | 0 | 0 | 1 | 1 | 1 | 1 | 1 | 9 |
| TC5 | 1 | 1 | 1 | 1 | 1 | 1 | 0 | 0 | 1 | 1 | 1 | 9 |
| TC10 | 1 | 1 | 1 | 1 | 1 | 0 | 0 | 1 | 1 | 1 | 1 | 9 |
| TC1 | 1 | 1 | 0 | 1 | 0 | 0 | 1 | 1 | 1 | 1 | 1 | 8 |
| TC3 | 1 | 1 | 1 | 0 | 0 | 1 | 1 | 1 | 1 | 0 | 1 | 8 |
| TC6 | 1 | 1 | 1 | 1 | 0 | 0 | 0 | 1 | 1 | 1 | 1 | 8 |
| TC7 | 1 | 1 | 1 | 1 | 1 | 0 | 0 | 0 | 1 | 1 | 1 | 8 |
| TC12 | 1 | 1 | 1 | 1 | 1 | 0 | 0 | 0 | 1 | 1 | 1 | 8 |
| TC13 | 1 | 1 | 1 | 1 | 1 | 1 | 1 | 1 | 0 | 0 | 0 | 8 |
| TC15 | 1 | 1 | 1 | 1 | 1 | 1 | 1 | 1 | 0 | 0 | 0 | 8 |
| TC8 | 0 | 0 | 1 | 0 | 1 | 1 | 1 | 0 | 1 | 1 | 1 | 7 |
| TC14 | 1 | 1 | 1 | 0 | 1 | 1 | 1 | 1 | 0 | 0 | 0 | 7 |
| TC2 | 1 | 1 | 0 | 0 | 0 | 1 | 1 | 1 | 1 | 0 | 0 | 6 |

Table 7, the column-wise total, and Table 8, the columns arranged in descending order according to the total. Table 9 shows the formation of a dense cluster. The reason behind using such a cluster denotes one famous ideology for the research work. In Table 9, I

drew out four dense clusters available in different colors. In the yellow color dense cluster features like C1, C2, C3, C4 and C9 is 100% needed for the test cases TC9, TC11, TC4, TC5 and TC10. Features like C8, C11, C5, C7 and C10 are 100% for the test cases TC9 and TC11 in the violet color clusters. Out of thirty 1's, the red color cluster contains, there you can find only thirty-one 1's are available, that is 88.57%, features like C1, C2, C3, C9, and C4 are needed for the test cases TC1, TC3, TC6, TC7, TC12, TC13 and TC15. Similarly, for the naive green color cluster. The above is the new ideology for the clustering method in test cases for software application development.

**Table 7.** Column-wise total.

| TC | C1 | C2 | C3 | C9 | C4 | C8 | C11 | C5 | C7 | C10 | C6 | Row Total |
|---|---|---|---|---|---|---|---|---|---|---|---|---|
| TC9 | 1 | 1 | 1 | 1 | 1 | 1 | 1 | 1 | 1 | 1 | 0 | 10 |
| TC11 | 1 | 1 | 1 | 1 | 1 | 1 | 1 | 1 | 1 | 1 | 0 | 10 |
| TC4 | 1 | 1 | 1 | 1 | 1 | 1 | 1 | 0 | 1 | 1 | 0 | 9 |
| TC5 | 1 | 1 | 1 | 1 | 1 | 0 | 1 | 1 | 0 | 1 | 1 | 9 |
| TC10 | 1 | 1 | 1 | 1 | 1 | 1 | 1 | 1 | 0 | 1 | 0 | 9 |
| TC1 | 1 | 1 | 0 | 1 | 1 | 1 | 1 | 0 | 1 | 1 | 0 | 8 |
| TC3 | 1 | 1 | 1 | 1 | 0 | 1 | 1 | 0 | 1 | 0 | 1 | 8 |
| TC6 | 1 | 1 | 1 | 1 | 1 | 1 | 1 | 0 | 0 | 1 | 0 | 8 |
| TC7 | 1 | 1 | 1 | 1 | 1 | 0 | 1 | 1 | 0 | 1 | 0 | 8 |
| TC12 | 1 | 1 | 1 | 1 | 1 | 0 | 1 | 1 | 0 | 1 | 0 | 8 |
| TC13 | 1 | 1 | 1 | 0 | 1 | 1 | 0 | 1 | 1 | 0 | 1 | 8 |
| TC15 | 1 | 1 | 1 | 0 | 1 | 1 | 0 | 1 | 1 | 0 | 1 | 8 |
| TC8 | 0 | 0 | 1 | 1 | 0 | 0 | 1 | 1 | 1 | 1 | 1 | 7 |
| TC14 | 1 | 1 | 1 | 0 | 0 | 1 | 0 | 1 | 1 | 0 | 1 | 7 |
| TC2 | 1 | 1 | 0 | 1 | 0 | 1 | 0 | 0 | 1 | 0 | 1 | 6 |
| Col-Total | 14 | 14 | 13 | 12 | 11 | 11 | 11 | 10 | 10 | 10 | 7 | |

**Table 8.** Ranking made according to the column-wise total in descending order.

| TC | C1 | C2 | C3 | C4 | C5 | C6 | C7 | C8 | C9 | C10 | C11 | Row Total |
|---|---|---|---|---|---|---|---|---|---|---|---|---|
| TC9 | 1 | 1 | 1 | 1 | 1 | 0 | 1 | 1 | 1 | 1 | 1 | 10 |
| TC11 | 1 | 1 | 1 | 1 | 1 | 0 | 1 | 1 | 1 | 1 | 1 | 10 |
| TC4 | 1 | 1 | 1 | 1 | 0 | 0 | 1 | 1 | 1 | 1 | 1 | 9 |
| TC5 | 1 | 1 | 1 | 1 | 1 | 1 | 0 | 0 | 1 | 1 | 1 | 9 |
| TC10 | 1 | 1 | 1 | 1 | 1 | 0 | 0 | 1 | 1 | 1 | 1 | 9 |
| TC1 | 1 | 1 | 0 | 1 | 0 | 0 | 1 | 1 | 1 | 1 | 1 | 8 |
| TC3 | 1 | 1 | 1 | 0 | 0 | 1 | 1 | 1 | 1 | 0 | 1 | 8 |
| TC6 | 1 | 1 | 1 | 1 | 0 | 0 | 0 | 1 | 1 | 1 | 1 | 8 |
| TC7 | 1 | 1 | 1 | 1 | 1 | 0 | 0 | 0 | 1 | 1 | 1 | 8 |
| TC12 | 1 | 1 | 1 | 1 | 1 | 0 | 0 | 0 | 1 | 1 | 1 | 8 |
| TC13 | 1 | 1 | 1 | 1 | 1 | 1 | 1 | 1 | 0 | 0 | 0 | 8 |
| TC15 | 1 | 1 | 1 | 1 | 1 | 1 | 1 | 1 | 0 | 0 | 0 | 8 |
| TC8 | 0 | 0 | 1 | 0 | 1 | 1 | 1 | 0 | 1 | 1 | 1 | 7 |
| TC14 | 1 | 1 | 1 | 0 | 1 | 1 | 1 | 1 | 0 | 0 | 0 | 7 |
| TC2 | 1 | 1 | 0 | 0 | 0 | 1 | 1 | 1 | 1 | 0 | 0 | 6 |
| Col-Total | 14 | 14 | 13 | 11 | 10 | 7 | 10 | 11 | 12 | 10 | 11 | |

**Table 9.** Denser cluster formed.

| TC | C1 | C2 | C3 | C9 | C4 | C8 | C11 | C5 | C7 | C10 | C6 | Row Total |
|---|---|---|---|---|---|---|---|---|---|---|---|---|
| TC9 | 1 | 1 | 1 | 1 | 1 | 1 | 1 | 1 | 1 | 1 | 0 | 10 |
| TC11 | 1 | 1 | 1 | 1 | 1 | 1 | 1 | 1 | 1 | 1 | 0 | 10 |
| TC4 | 1 | 1 | 1 | 1 | 1 | 1 | 1 | 0 | 1 | 1 | 0 | 9 |
| TC5 | 1 | 1 | 1 | 1 | 1 | 0 | 1 | 1 | 0 | 1 | 1 | 9 |
| TC10 | 1 | 1 | 1 | 1 | 1 | 1 | 1 | 1 | 0 | 1 | 0 | 9 |
| TC1 | 1 | 1 | 0 | 1 | 1 | 1 | 1 | 0 | 1 | 1 | 0 | 8 |
| TC3 | 1 | 1 | 1 | 1 | 0 | 1 | 1 | 0 | 1 | 0 | 1 | 8 |
| TC6 | 1 | 1 | 1 | 1 | 1 | 1 | 1 | 0 | 0 | 1 | 0 | 8 |
| TC7 | 1 | 1 | 1 | 1 | 1 | 0 | 1 | 1 | 0 | 1 | 0 | 8 |
| TC12 | 1 | 1 | 1 | 1 | 1 | 0 | 1 | 1 | 0 | 1 | 0 | 8 |
| TC13 | 1 | 1 | 1 | 0 | 1 | 1 | 0 | 1 | 1 | 0 | 1 | 8 |

**Table 9.** *Cont.*

| TC | C1 | C2 | C3 | C9 | C4 | C8 | C11 | C5 | C7 | C10 | C6 | Row Total |
|----|----|----|----|----|----|----|-----|----|----|-----|----|-----------|
| TC15 | 1 | 1 | 1 | 0 | 1 | 1 | 0 | 1 | 1 | 0 | 1 | 8 |
| TC8 | 0 | 0 | 1 | 1 | 0 | 0 | 1 | 1 | 1 | 1 | 1 | 7 |
| TC14 | 1 | 1 | 1 | 0 | 0 | 1 | 0 | 1 | 1 | 0 | 1 | 7 |
| TC2 | 1 | 1 | 0 | 1 | 0 | 1 | 0 | 0 | 1 | 0 | 1 | 6 |
| Col-Total | 14 | 14 | 13 | 12 | 11 | 11 | 11 | 10 | 10 | 10 | 7 | |

### 3.1. Methodology 1

The first approach for test case selection uses the rank order clustering methodology. The algorithm is given below:

Rank order clustering algorithm:

Step 1: Convert the ordinal values to cardinal values.

Step 2: Assign binary weight and determine the total weight for each row, say

$$TW_i = \sum_{i=1}^{m} b_{ij} 2^{m-j} \tag{1}$$

where $m$ is the features, $i$ is the number of rows. $b_{ip}$ has a value between 1 and 5, depending on the matrix.

The binary number for test cases($n$) and features($m$) is denoted as $b_{ip}$. Where, $b_{ip}$ is simply a $n * m$ matrix.

Step 3: Reorder the rows in the descending order of $TW_i$ values.

Step 4: Assign binary weight and determine the total weight for each column, say $TW_j$.

The expression can be understood as follows:

$$TW_j = \sum_{j=1}^{n} b_{ji} 2^{n-i} \tag{2}$$

where $n$ is the total number of test cases, $j$ is the number of columns.

Step 5: Reorder the columns in the descending order of $TW_j$ values.

Step 6: Fix the threshold value.

Step 7: Cluster (CR) initialized with zero DataPoints.

    Step 7.1: Mark DataPoint (DP) with value 1 as visited.

    Step 7.2: Find NeighborsSet (NS) of the selected DataPoint (DP)

    Step 7.3: If (NeighborsSet (NS) has more than three 1's) AND If (DataPoint (DP) is not yet part of any cluster (CR)).

    Step 7.4: DataPoints (DP) selected according to the requirement.

    Step 7.5: Add DataPoint (DP) to Cluster (CR).

    Step 7.6: For each unvisited DataPoint in NeighborsSet, Perform from Step

    Step 7.7: Else,

    Step 7.8: Mark DataPoint as NoisePoint.

### 3.2. Initializing the Agent for the Proposed Algorithm

1.  The next step is calculating the similarity or rank order between the data points.
2.  The Rank Order Clustering algorithm agents represent the initial cluster centers or centroids. These agents need to be initialized to start the clustering process.
3.  Once the agents are initialized, the algorithm proceeds with an iterative process to assign data points to clusters based on their similarity to the agents' centroids.
4.  After the clustering process converges, the quality of the resulting clusters is evaluated using appropriate metrics, such as within-cluster sum of squares or other domain-specific performance measures.

### 3.3. Algorithm 1. Rank Ordering and Clustering

The application of the above method for education is explained in Table 2 as follows: The test problem in Table 2 is prepared according to the number of connected values. The

properties are listed in chronological order. These attributes are given coefficients of $2^0$, $2^1$, $2^2$ ... $2^m$ from right to left. The cost of training for each test is equal to the coefficient of behavior and income. Then, the tests are sorted by the number of values. In the next step, serial totals are calculated on a column-by-column basis for the test data set for the previous step's results. These tests gave coefficients of $2^0$, $2^1$, $2^2$ ... $2^m$ from bottom to top. The training value for each attribute is multiplied by the coefficient of the relevant data to get the number. The product is then rearranged to the lower part of that price. In our method, the default position is 3. Therefore, more than three values are considered 1 and smaller values are considered 0. Table 3 is converted to the binary matrix and groups are formed according to the written process. This process looks like this in Table 4 for our training process: The table is clustered with intensities 1s using Algorithm 1. The following information can be taken into account from the table above: Factors 15, 12, and 5 when focusing on critical illness, needs, time, weak needs, failure proneness, cover LOC, complexity, and detection of errors when using functions such as complexity, 13, 9, 11 and 7, recovery can be selected with 100% confidence. When focusing on performance (e.g., critical illness, required coverage, and duration), test data including 3, 1, 6, 2, 14, 10, 4, and 8 can be selected for repeated measures with a confidence level of 83%. While focusing on performance (i.e., need for change, failure rate, and LOC of coverage), test takers 3, 1, 6, 2, 14, 10, 4, and 8 can be safely selected for repeated measures at the 75% level. When focusing on performance (i.e., using complexity, length, and customer importance), tests 3, 1, 6, and 2 can be selected for regression analysis with a confidence level of 75%. Chapter Consider weight to score.

Weights are 1, 2, 4, 16, etc., in the workshop from right to left and from bottom to top. Therefore, Table 2 is the base table for collecting the series. The first step is to sort the training in the paragraph below. Then, count the rows line by line and sort them in the following order. The resulting matrix is then converted to a table containing only binary values. Binary conversion is conducted using the threshold of 3. Values greater than 3 will be converted to 1, and values less than 3 will be converted to 0. Because of this conversion, the results in Table 3 are heavily positive with 1s. Dense 1s are grouped according to the algorithm to be grouped. A group of test cases contain a certain percentage of the characteristics identified in that group. Therefore, these tests can be selected from the set of test data available to validate specific functions. This is a technological know-how in case selection. This method is more useful in the selection of retrospective tests.

### 3.4. Methodology 2—Fuzzy-Based Similarity Coefficient-Based Clustering

In this methodology also, test cases along with the 11 different features are considered. These features are the same set of attributes that were used in Methodology 1. The inputs for this approach are purely binary values. These binary values are obtained by converting the matrix values ranging from 1 to 5 into binary values by keeping a threshold value. The values equal to or greater than the threshold value are 1, and the remaining are 0s. Hence, the test case values in Table 3 are transformed into binary based on a threshold value of 3. So, Table 3 is converted to Table 5 as given below:

As a next step, the similarity among test cases concerning 11 features is found using the following formula:

$$SR_{ij} = \mathbf{NPCM_{ij}} + \mathbf{NPMI_i} + \mathbf{NPMJ_j} - \mathbf{NPCM_{ij}} \tag{3}$$

where, $\mathbf{NPCM_{ij}}$ = number of common 1s in both rows i and j, $\mathbf{NPMI_i}$ = number of 1s in row 1. $\mathbf{NPMJ_j}$ = number of 1s in row j.

For the values in Table 10, the similarity coefficient values are found using the Formula (4). The similarity coefficient values for the training set values are formed in Table 6, as given below:

**Table 10.** Cosine similarity coefficient.

| TC | TC1 | TC2 | TC3 | TC4 | TC5 | TC6 | TC7 | TC8 | TC9 | TC10 | TC11 | TC12 | TC13 | TC14 | TC15 |
|---|---|---|---|---|---|---|---|---|---|---|---|---|---|---|---|
| TC1 | 1 | 0.8 | 0.8 | 0.9 | 0.9 | 0.9 | 0.9 | 0. 6 | 0.9 | 0.9 | 0.9 | 0.9 | 0.9 | 0.9 | 0.9 |
| TC2 | 0.8 | 1 | 0.8 | 0.8 | 0.8 | 0.8 | 0.8 | 0. 6 | 0.8 | 0.8 | 0.8 | 0.8 | 0.8 | 0.8 | 0.8 |
| TC3 | 0.8 | 0.8 | 1 | 0.8 | 0.8 | 0.8 | 0.8 | 0. 6 | 0.8 | 0.8 | 0.8 | 0.8 | 0.8 | 0.8 | 0.8 |
| TC4 | 0.9 | 0.8 | 0.8 | 1 | 0.9 | 0.9 | 0.9 | 0. 6 | 0.9 | 0.9 | 0.9 | 0.9 | 0.9 | 0.9 | 0.9 |
| TC5 | 0.9 | 0.8 | 0.8 | 0.9 | 1 | 0.9 | 0.9 | 0. 6 | 0.9 | 0.9 | 0.9 | 0.9 | 0.9 | 0.9 | 0.9 |
| TC6 | 0.9 | 0.8 | 0.8 | 0.9 | 0.9 | 1 | 1 | 0. 6 | 0.9 | 0.9 | 0.9 | 1 | 1 | 0.9 | 1 |
| TC7 | 0.9 | 0.8 | 0.8 | 0.9 | 0.9 | 1 | 1 | 0. 6 | 0.9 | 0.9 | 0.9 | 1 | 1 | 0.9 | 1 |
| TC11 | 0.9 | 0.8 | 0.8 | 0.9 | 0.9 | 0.9 | 0.9 | 0. 6 | 1 | 0.9 | 1 | 0.9 | 0.9 | 0.9 | 0.9 |
| TC12 | 0.9 | 0.8 | 0.8 | 0.9 | 0.9 | 1 | 1 | 0. 6 | 0.9 | 0.9 | 0.9 | 1 | 1 | 0.9 | 1 |
| TC13 | 0.8 | 0.8 | 0.8 | 0.8 | | 0.8 | 0.9 | 0.6 | 0.8 | 0.8 | 0.9 | 0.9 | 0.8 | 0.8 | 0.9 |
| TC14 | 0.9 | 0.8 | 0.8 | 0.9 | 0.9 | 0.9 | 0.9 | 0. 6 | 0.9 | 0.9 | 0.9 | 0.9 | 0.9 | 1 | 0.9 |
| TC15 | 0.9 | 0.8 | 0.8 | 0.9 | 0.9 | 1 | 1 | 0. 6 | 0.9 | 0.9 | 0.9 | 1 | 1 | 0.9 | 1 |

The above fuzzy relation on a single universe TC is also a relation from TC to TC. This will become fuzzy equivalence when it satisfies the following three properties:

$$\text{Reflexivity}: \ \mu\widetilde{FR}\left(TC_i,\ TC_j\right) = \mu\widetilde{FR}\left(TC_j,\ TC_j\right) \tag{4}$$

$$\text{Symmetry}: \ \mu\widetilde{FR}\left(TC_i,\ TC_j\right) = \mu\widetilde{FR}\left(TC_j,\ TC_i\right) \tag{5}$$

$$\text{Transitivity}: \ \mu\widetilde{FR}\left(TC_i,\ TC_j\right) = \lambda_1 \text{ and } \mu\widetilde{FR}\left(TC_j,\ TC_k\right) = \lambda_2, \tag{6}$$

then $\mu\widetilde{FR}(TC_i,\ TC_k) = \lambda_3$, where $\lambda \geq \min[\lambda_1, \lambda_2, \lambda_3]$ and $\mu\widetilde{FR}$ is the membership fuzzy relation.

In fuzzy set theory, a fuzzy relation is reflexive if every element of the set is related to itself with a degree of membership equal to 1. Mathematically, a fuzzy relation R on a fuzzy set F is reflexive if: This means that the membership degree of each element $TC_i$ in F concerning itself is equal to 1. Reflexivity in the context of $\mu\widetilde{FR}\left(TC_i,\ TC_j\right) = \mu\widetilde{FR}(TC_i,\ TC_j)$, implies that there is a strong correlation or similarity between two entities $TC_i$ and $TC_j$. The software application's capacitance conversion functionality for membership fuzzy ($\mu F$) to fuzzy (F) and vice versa has been tested and is functioning correctly. The application accurately converts values within the valid range and handles invalid input gracefully.

Where $\lambda_1$ and $\lambda_2$ are the membership degrees of the fuzzy relations between elements $TC_i$ and $TC_j$ and between elements $TC_i$ and $TC_k$, respectively. Then, transitivity requires that $\lambda \geq \min[\lambda_1, \lambda_2, \lambda_3]$, indicating that the membership degree between $TC_i$ and $TC_j$ should be greater than or equal to the minimum of the membership degrees between $TC_i$ and $TC_j$, and $TC_i$ and $TC_k$.

The maxmin composition must be performed until the relation attains transitivity to transform the tolerance into an equivalence relation. The number of times the max–min compositions can be applied will be (n–), where n represents the number of elements in the relation.

The fuzzy relation in Table 6 is the fuzzy tolerance relation. His should be converted into a fuzzy equivalence relation. The fuzzy equivalence relation has properties: reflexivity, symmetry, and transitivity. Already, the values in Table 6 possess reflexivity and symmetry properties. To achieve transitivity, applying fuzzy maxmin composition is necessary. Hence, applying fuzzy maxmin compositions for achieving transitivity is performed, and the result is given below:

Clusters are formed with test cases having the same similarity scores. These clusters show the degree of strength in the relationship among test cases. As a result, it is enough to choose one test case per cluster as these are similar to a certain degree. And highly dissimilar test cases are ignored, as it establishes noise. Ultimately, a few test cases are not selected. The remaining selected cases yield coverage among all the 11 considered features,

which serves the purpose. From Table 8, the TC4 can be selected by ignoring the test cases TC5, TC6, TC7, TC9, TC10, TC11, TC12, TC13, TC14, and TC15 since they are equivalent. Hence, the test cases selected for a test suite at a 0.9 confidence level would be TC1, TC2, TC3, TC4, and TC8.

Similarly, test cases can be selected for a test suite at various confidence levels. The challenge of not compromising on quality is achieved because of equivalence classification. Another significance of this method is all 11 features are covered with the help of selected test cases.

*3.5. Test Case Functionality for E-Commerce Application*

Test Case Title: User Registration with Valid Data.

Test Case Description:

This test case verifies the functionality of the user registration process in the e-commerce application, specifically focusing on registering a new user with valid data. The objective is to ensure that the system accurately processes and accepts valid user input during registration.

Preconditions:

1.　　The e-commerce application is accessible and operational.
2.　　The user is on the registration page.

Test Steps:

1.　　Launch the e-commerce application.
2.　　Navigate to the registration page.
3.　　Enter valid user data in the registration form:

    a.　　Full Name: enter a valid full name (e.g., John Doe).
    b.　　Email Address: Enter a valid email address in the correct format (e.g., john.doe@example.com).
    c.　　Password: enter a valid password that meets the specified criteria (e.g., at least 8 characters long with a mix of uppercase, lowercase, and numbers).
    d.　　Confirm password: re-enter the same valid password to confirm.
    e.　　Phone number: enter a valid phone number (e.g., 1234567890) in the correct format.
    f.　　Address: enter a valid address (e.g., 1234 Street Name, City, State, Zip Code).

4.　　Click the "Register" or "Sign Up" button.
5.　　Verify that the system accepts the provided data and successfully registers the user.
6.　　Check for a confirmation message or email indicating successful registration.
7.　　Log in with the registered credentials to confirm successful registration.

Expected Result:

The system should accept the valid user data in the registration form and successfully register the user. A confirmation message or email should be received, and the user should be able to log in using the registered credentials.

Pass/Fail Criteria:

—　　Pass: the system accepts the valid data, registers the user successfully, and provides confirmation of the registration.
—　　Fail: the system does not accept the valid data or encounters errors during the registration process, preventing successful registration.

A crucial aspect of creating effective test cases is to include all essential information required for successful execution. For instance, when verifying the login functionality on a website, the test case must encompass both the login and password details. Without these credentials, the test case would be insufficient and unable to achieve the desired testing objectives. In summary, comprehensive and accurate information within each test case is pivotal for their effectiveness and ensuring the intended testing goals are met, in this case, authenticating login functionality on the site.

*3.6. Test Case for E-Commerce Application*

Test Case Identifier: TC_ECOM_001.
Title: Adding a Product to the Cart.
Description: Verify the functionality of adding a product to the cart in the e-commerce application.
Preconditions:
User is registered and logged in.
The application is accessible and properly functioning.
Steps:
Open the e-commerce application.
Log in with valid credentials.
Browse the product catalog and select a product to add to the cart.
Click on the product to view its details.
Locate the "Add to Cart" button and click on it.
Verify the product is added to the cart.
Expected Outcome:
The application successfully adds the selected product to the cart.
The cart displays the product added with the correct details and quantity.

In an effective test case, certain elements should be avoided to ensure clarity and accuracy. Firstly, dependencies on other test cases should be minimized, as each test case should be standalone to facilitate easy execution and interpretation. Secondly, unclear formulation of steps or expected results hamper the test's purpose and effectiveness, making it essential to maintain clear and precise instructions. Thirdly, including all necessary information is crucial for a test case to be passable and meaningful. Lastly, excessive detail should be avoided to maintain focus and relevance, ensuring that the test case remains concise and to the point, enhancing efficiency in testing processes. These principles promote well-structured and effective test cases that contribute to robust software testing practices.

## 4. Empirical Evaluation and Metrics

The efficiency of the implemented methodology needs to be measured, without which the research would not be complete. Hence, two metrics have been proposed as stated below:

The Selection metric is used to find the savings in effort, which is as given below:

$$PSR = \{TCS \times AS\} \times 100 \tag{7}$$

PSR stands for percentage of selection reduced; TCS denotes selected test cases; As means selected attributes; and AS signifies a total number of attributes.

The efficiency metric given below is to find the efficiency of test cases in detecting defects:

$$\textbf{Test efficiency} = \frac{D}{N} \times 100 \tag{8}$$

where D denotes defects covered, and N denotes the number of test cases.

$$\textbf{Percentage Decrease} = \frac{\textbf{Starting Value} - \textbf{Ending Value}}{|\textbf{Starting Value}|} \times 100 \tag{9}$$

where:

"Starting Value" is the initial value or the value before the change.
"Ending Value" is the final value or the value after the change.

The methodologies have been implemented and verified with five different software systems for the experimental study. These software systems have been developed using the NET framework and are available as running software systems. The intricacies of these software applications are shown in Table 9.

## 5. Results and Discussion

The experiments were performed rigorously with the available test cases mentioned in Table 4. The results of the experimental study are detailed in Table 4. As per the results available in Table 4, selected test cases cover the same number of defects. The cosine similarity coefficient is a measure of similarity between two non-zero vectors defined in an inner product space. Cosine similarity is its low complexity, especially for sparse vectors, where only the non-zero coordinates must be considered. Hence, the quality is not at all compromised. In Table 10, similarity has been calculated using cosine similarity measures. TC1 is compared with TC1. It is one. TC1 with TC2, TC1 with TC3, TC1 with TC4 and so on up to TC1 with TC15. The same other rows are compared to these results, and they exhibited the same in Table 10. A value of 1 represents high similarity, and 0.9, 0.8, and 0.6 represent 90%, 80%, and 60%, respectively.

In Table 11, 1.0 value of test cases are clustered, 0.9 value of test cases are clustered, 0.8 value of test cases are clustered, and 0.0 value of test cases are clustered as such. Herewith, it represents some of the test cases that are 100% required, 90%, 80%, and 60% required according to the need of the software applications development. In Table 12, software packages are taken from different domains, and the corresponding attributed values are measured and the same in the table. In Table 13, results, methodologies, and measures are shown.

**Table 11.** Lambda cut.

| Lambda Cut | Test Cases Clustered |
|---|---|
| 1 | {TC1}, {TC2}, {TC3}, {TC4}, {TC5},{TC6, TC7, TC12, TC13, TC15}, {TC8}, |
| 2 | {TC9, TC11}, {TC10}, {TC14} |
| 0.9 | {TC4, TC5, TC6, TC7, TC9, TC10, TC11, TC12, TC13, TC14, TC15},{TC1}, {TC2}, {TC3}, {TC8} |
| 0.8 | {TC2, TC3, TC4, TC5, TC6, TC7, TC9, TC10, TC11, TC12, TC13, TC14, TC15},{TC1}, {TC8} |
| 0.6 | {TC2, TC3, TC4, TC5, TC6, TC7, TC8, TC9, TC10, TC11, TC12, TC13, TC14, TC15}, {TC1} |
| 0 | {TC1, TC2, TC3, TC4, TC5, TC6, TC7, TC8, TC9, TC10, TC11, TC12, TC13, TC14, TC15} |

**Table 12.** Software applications and test suits.

| S.No | Applications | Size in KLOC | No. of Modules | No. of Defects Covered | Test Suit Size |
|---|---|---|---|---|---|
| 1 | Railway reservation system | 16.4 | 105 | 1403 | 1912 |
| 2 | Hospital management system | 22.3 | 78 | 1231 | 1691 |
| 3 | Online banking system | 12.7 | 89 | 1601 | 2020 |
| 4 | College management system | 15 | 93 | 1172 | 1446 |
| 5 | Municipal tax portal | 22.3 | 110 | 2630 | 3050 |
| 6 | ERP Systems | 22.1 | 112 | 2336 | 2934 |
| 7 | CRM | 21.7 | 108 | 2145 | 2856 |

Through the second methodology, similarity index and fuzzy inference, the research noticed one good idea: to understand the amount of test case percentage needed to test the software application's real motive. In Figure 1, an efficiency chart shows the software application's real efficiency after being deployed into the testing. Everything has been calculated using Methodology 1 and Methodology 2. In Figure 2, the percentage of selection reduced chart was deployed using PSR (percentage of selection reduced) methods 1 and 2 using Equations (8) and (9).

**Table 13.** Results of methodologies and measures.

| S.No | Software Package | Test Cases Selected(TCs)-Methodology 1 | No. of Attributes Selected (As) out of 11-Methodology 1 | Efficiency-Methodology 1 | PSR-Methodology 1 | Test Cases Selected-Methodology 2 | Efficiency-Methodology 2 | PSR-Methodology 2 |
|---|---|---|---|---|---|---|---|---|
| 1 | Railway reservation system | 1700 | 8 | 82.5 | 62 | 1600 | 87.6 | 80.3 |
| 2 | Hospital management system | 1434 | 6 | 85.8 | 45.7 | 1333 | 92.3 | 78.8 |
| 3 | Online banking system | 1856 | 9 | 86.2 | 74.3 | 1798 | 89 | 89 |
| 4 | College management system | 1219 | 10 | 96.1 | 75.9 | 1273 | 92 | 88 |
| 5 | Municipal tax portal | 2833 | 6 | 92.8 | 50 | 2911 | 90.3 | 95.4 |
| 6 | ERP System | 2746 | 7 | 91.7 | 53 | 2813 | 91.3 | 93.4 |
| 7 | CRM | 2689 | 6 | 90.6 | 56 | 2798 | 90.4 | 92.3 |

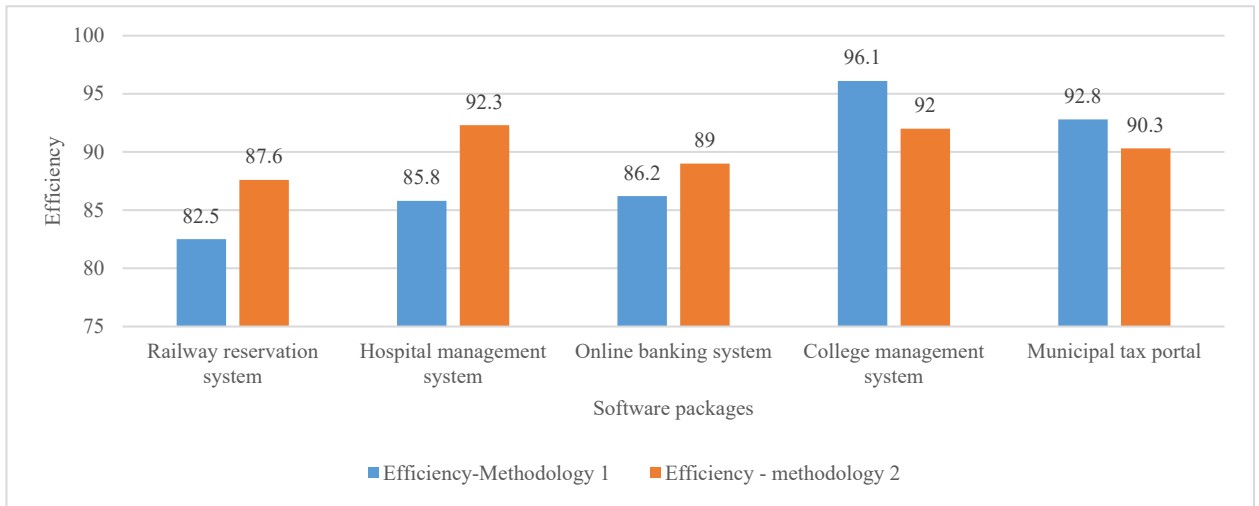

**Figure 1.** Efficiency chart.

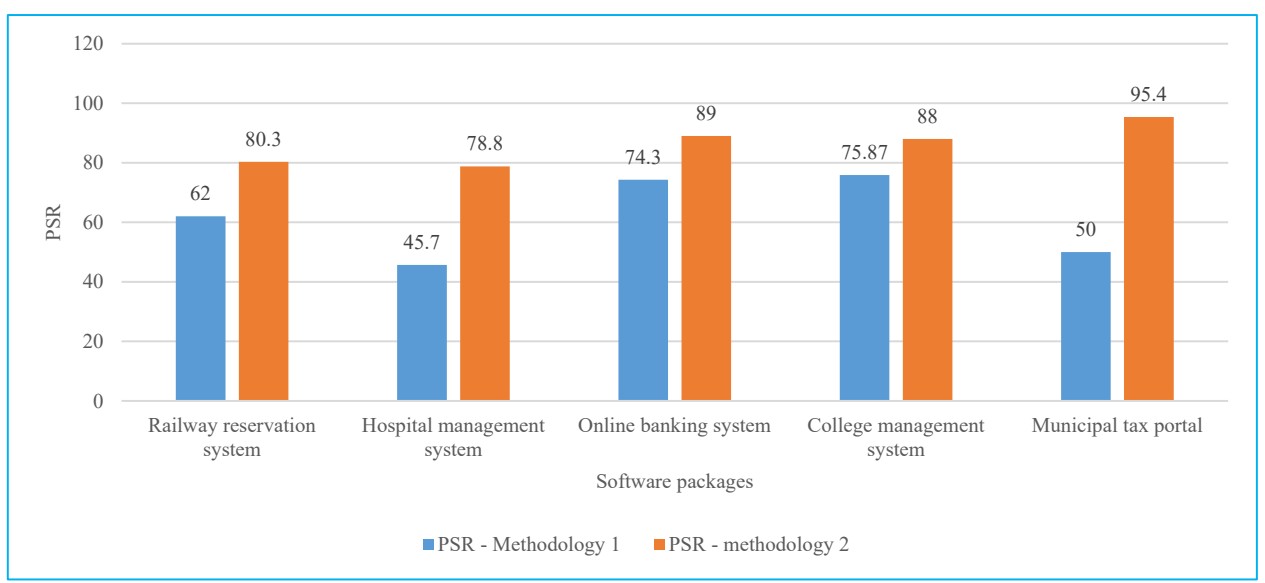

**Figure 2.** Percentage of selection reduced chart.

$$
\begin{aligned}
\text{Total Test Suit Size} \\
= \text{ Number of Equivalence Classes} \\
\times \text{ Number of Test Cases per Equivalence Class}
\end{aligned} \tag{10}
$$

$$
\begin{aligned}
\text{Total Test Suit Size} \\
= \text{ Number of Boundary Values} \\
\times \text{ Number of Test Cases per Boundary Values}
\end{aligned} \tag{11}
$$

Table 12 evaluates the values as per the mathematical equations of (10) and (11). All the results have been found through the Python implementations. The fuzzy similarity index and Lambda cut have been used to overcome the issue of testing time optimization, as shown in Tables 2–11. The undefined similarity index is a technique used for approximate string matching and is valuable for various data science tasks. On the other hand, "Lambda Cut" appears to be a concept related to fuzzy logic.

In the first methodology, test cases are selected for the attributes being focused. Whereas in the second methodology, all the considered attributes are covered. Also, the second methodology gives a higher reduction percentage among test cases. Both methodologies were found to be highly efficient in fault detection and reduction in test cases. Moreover, both methodologies have considered 11 attributes that were never used in the earlier research. Hence, regarding attribute coverage, efficiency, and percentage of selection reduced in test cases, the methods implemented have proved to be highly successful in selecting regression test cases.

In Table 13, all the measurements are taken according to the mathematical formula exhibited for reference by Equations (7) and (12).

$$
\textbf{Efficiency} = \frac{\textbf{Time CPU is busy}}{\textbf{Total Time}} * \textbf{100} \tag{12}
$$

In Table 14, the e-commerce application, each test case includes a test case ID, description, input data, expected outcome, and potential reductions in time, cost, and complexity. These reductions are hypothetical and should be based on a detailed analysis and evaluation of the application and the testing process.

**Table 14.** E-commerece application.

| Test Case ID | Test Case Description | Input Data | Expected Outcome | Time (Reduction) | Cost (Reduction) | Complexity (Reduction) |
|---|---|---|---|---|---|---|
| TC001 | User registration with valid data | User info | Successful registration | −10% | −15% | −5% |
| TC002 | User login with correct credentials | Credentials | Successful login | −5% | −10% | −5% |
| TC003 | Product search by name and category | Search data | Accurate product search results | −15% | −10% | −10% |
| TC004 | Add product to cart and proceed to checkout | Product ID | Seamless transition to checkout page | −10% | −5% | −5% |
| TC005 | Payment process with valid payment details | Payment info | Successful payment | −10% | −10% | −10% |
| TC006 | User profile update with valid information | Profile info | Profile updated successfully | −5% | −5% | −5% |
| TC007 | User account deletion with confirmation | N/A | Account deleted successfully | −10% | −10% | −5% |
| TC008 | Product recommendation based on browsing history | User history | Accurate product recommendations | −10% | −10% | −5% |
| TC009 | Checkout process with multiple items in the cart | Cart items | Seamless checkout for multiple items | −15% | −10% | −10% |
| TC010 | Mobile responsiveness across various devices | N/A | Consistent UI/UX on different devices | −20% | −20% | −15% |

The acronym PSR represents "Priority to Severity Ratio", a notable metric employed within software testing. This metric is pivotal in establishing the optimal sequence for

addressing defects or issues encountered during testing phases. Its chief utility is enabling teams to allocate their resources judiciously, considering two vital aspects: the severity of a particular defect and its corresponding priority. The underlying formula used to compute the PSR value is articulated as follows:

*Analysis of the Result*

The research study results provide useful results for improving error detection in a short time. Applying competitive analysis using the ranking and similarity coefficient can improve the method's performance. The results from the experiment show that there has been a significant improvement in the detection of errors, and thus, the testing process has been improved. This advancement is essential for software development, where time and resources are limited to optimize test suites. An integrated grey-box testing approach allows software developers to simplify testing while providing comprehensive coverage. As other studies in this area have shown, empirical results are based on the necessity of selection to obtain good results and the importance of empirical problems in the sources. Sorting is ordering data points according to a particular measure or measure. In production process analysis, sort clustering is an algorithm for classifying machines according to their needs for different products. The steps involved in the group analysis are as follows: For each row (item), calculate the number representing the activity of each machine.

Select the following line as the count number.

For each column (machine), calculate the number representing the product that needs to be processed on that machine.

Check the columns in order by the number of numbers.

If there is no change in steps 2 and 4, stop; otherwise, repeat the process.

The final result is the ranking of machines according to their importance in processing different products.

Clustering or cluster analysis is a method for grouping data points based on their similarities.

In traditional integration, data points are assigned to different groups, and each data points to only one group. Clusters can be defined using different metrics such as distance, connectivity, and power. The aim is to maximize similarity within the group while minimizing similarity between different groups. Fuzzy clustering or soft k-means is a form of clustering where each data point can belong to more than one cluster. Unlike traditional clustering, where clustering of data points is difficult, fuzzy clustering assigns membership levels to data points and indicates how often each group belongs. The fuzzy C-means (FCM) algorithm is a widely used fuzzy clustering algorithm. It involves assigning coefficients to each group's data points, calculating each group's centroid, and re-adjusting the membership coefficients until they converge. Lambda Slice is a fundamental technique that turns fuzzy membership into an intelligent classification. It helps to convert fuzzy members in binary assignments. Choosing an appropriate lambda value means assigning data points with membership levels higher than the threshold to a particular group. Instead, those below the threshold are not included in the group.

Research in bioinformatics has also explored the use of fuzzy-based clustering techniques to combine multivariate data. Based on fuzzy logic, fuzzy equivalence relations, and fuzzy similarity of Łukasiewicz values, the FH-Clust method has been proposed to identify subsets of patients from different omics data such as gene expression, miRNA expression, and methylation. This approach focuses on integrating patient data from multiple omics sites using a consensus matrix as analysis of single omics data ultimately leads to better results and greater clinical relevance. In Table 14, comparative analysis is an essential part of software testing. It involves comparing the actual output of a program or system with the expected output to ensure that it functions correctly.

Table 15, the proposed model, utilizing a measure of closeness, predictive modeling, and handling imprecise data, excels in quantifying resemblance and predicting outcomes based on variables. Unlike traditional models, it effectively navigates data scale challenges

and offers superior predictive capabilities, making it an innovative and versatile choice for diverse applications.

**Table 15.** Comparative analysis of models.

| Model Name | Key Features | Advantages | Limitations |
|---|---|---|---|
| Finite State Models | Represents system behavior using finite states and transitions | - Effective for systems with well-defined states and transitions | - Limited to systems that can be accurately represented as finite states |
| Linear Equation | - Uses linear equations to model relationships between variables | - Mathematically well-understood and widely applicable | - Assumes linear relationships, which may not always be accurate |
| Path Analysis | - Analyzes potential paths through a system or process | - Provides insights into potential scenarios and outcomes | - Can be complex for large and intricate systems |
| Dependency Graph | - Represents dependencies between components or elements | - Clearly visualizes relationships and dependencies | - Requires accurate and complete information on dependencies |
| Optimal Representative Set and Optimal Path Set Selection | - Select representative test cases or paths for efficient testing | - Reduces redundancy in test case selection, saving time and resources | - May not always capture all relevant test scenarios |
| Prediction Model Based on Coverage | - Predicts test coverage based on historical data or metrics | - Helps in prioritizing test cases for maximum coverage | - Accuracy of predictions can vary based on the quality of historical data |
| Data Mining for Test Case Selection | - Uses data mining techniques to identify relevant test cases | - Can discover hidden patterns and relationships in the data | - Requires a comprehensive and accurate dataset for effective mining |
| Dynamic Programming for Optimal Test Cases | - Finds optimal test cases using dynamic programming techniques | - Guarantees an optimal solution based on defined criteria | - Computationally intensive for large test case sets |
| Integer Linear Programming for Selection | - Utilizes integer linear programming to select optimal test cases | - Provides a rigorous and optimal selection process | - Requires expertise in mathematical modeling and solving ILP problems |
| Similarity-Based Test Selection Algorithm | - Selects test cases based on similarity to previous test outcomes | - Efficient for regression testing and reducing redundancy | - Requires a reliable similarity metric and historical test data |
| Fuzzy Expert System Based on Features | - Utilizes fuzzy logic and expert knowledge for test case selection | - Handles imprecise or uncertain information effectively | - Requires expertise in fuzzy logic and domain knowledge |
| Code Coverage Testing | - Measures the proportion of code exercised by test cases | - Helps in identifying untested parts of the code | - Does not guarantee thorough testing of all functionalities |
| Multi-Objective Evolutionary Algorithm | - Uses evolutionary algorithms to optimize multiple objectives in test case selection | - Balances multiple criteria for efficient test case selection | - May be computationally intensive for complex optimization problems |
| Optimal Control Based on Dependence Graph | - Utilizes optimal control theory and dependence graphs for efficient test case selection | - Incorporates system dependencies for optimized testing | - Requires a comprehensive understanding of the system and dependencies |
| Propose Model | Measure of closeness, predicts outcomes based on variables, represents uncertainty | Quantifies resemblance, predictive modeling, handles imprecise data | Sensitive to data scale, assumes linear relationship, complex interpretation |

## 6. Research Work Limitations

The proposed SCARF-RT model, while promising, has certain limitations. Firstly, the effectiveness heavily relies on selecting and weighing the eleven features, which could be subjective and may not generalize well across diverse software projects. The clustering techniques might not always yield optimal groupings, potentially leading to suboptimal test case selection. The research could benefit from a more comprehensive evaluation

across a broader range of software types and sizes to validate its efficacy in various contexts. Furthermore, scalability concerns may arise when dealing with large-scale software systems, necessitating further investigation for practical application.

*Instructions for Testing Everything*

In justifying the specific features of the e-commerce software application, it is crucial to highlight its comprehensive functionality and purpose within the testing context. The e-commerce application is a sophisticated software tailored for facilitating online buying and selling transactions, managing product inventories, processing payments securely, providing a seamless user interface, ensuring data privacy, integrating with various payment gateways, offering a personalized user experience through recommendations and preferences, enabling order tracking and customer support, and maintaining robust security measures against cyber threats. These features collectively contribute to the software's complexity and necessitate a tailored approach to testing, debunking the claim of being universally testable for "everything" due to its specialized functionalities and intricate design.

## 7. Conclusions and Future work

The research proposes a cutting-edge hybrid grey-box testing approach that combines the strengths of white-box and black-box techniques. By incorporating code flow and data flow analysis, our approach provides a transparent view of the software's internal workings, enabling efficient bug detection. This bridging of the gap between design models and actual code makes our approach a valuable tool for software testing. Additionally, we introduce a feature-based test case selection methodology that focuses on identifying the most critical aspects of the software. By involving skilled testers from various software industries, we establish a renowned order of importance for these features, independent of explicit or implicit requirements. This methodology can be applied to any test approach, resulting in an optimal set of test cases that effectively cover the most significant aspects of the software.

Furthermore, our research presents innovative quantum-based techniques for test case selection, leveraging the concept of quantum in feature coverage. Our approach optimizes the selection process and improves software testing effectiveness by establishing relationships between features and test cases through rank-based coefficients. These techniques specifically address the challenges of regression testing, allowing for the efficient identification of critical and showstopper errors within a minimal time. Finally, our proposed hybrid grey-box testing approach, combined with the feature-based test case selection methodology and quantum-based techniques, offers a comprehensive and advanced solution for software testing. By incorporating the strengths of different approaches and addressing critical aspects of the software, our research enhances the overall quality and reliability of software systems. The techniques used in the implemented approach are rank order and similarity coefficient-based clustering. Both methods have been applied to regression test selection, which is a very novel idea. A wide range of 11 features have been considered for the first time in test case selection. These techniques have been tested with several test suites, and it is found that the methodologies gave a satisfactory and convenient way of selecting test cases. Hence, undoubtedly, regression test efficiency is being improved by using these techniques. In the research approach, time, cost, and complexity reduction are very much a concern. Tables 12 and 13 exhibit the real outcome of the product rather than inventory. The work implemented, too, has some limitations. In method 2, clustering is achieved at a particular λ cut confidence level. Hence, the cluster sizes will vary according to λ cut level. Setting up this confidence level at a reasonable rate using the fuzzy technique, rather than making it a subjective decision, is our next focus of work. In method 1, some features were omitted during test case selection. Rather, complete coverage of all features needs to be attained. Also, the formation of clusters might vary based on the number of clusters chosen and the size of the cluster decided. For this purpose, clustering needs

improvisation, yet another future research work. Moreover, we have used only two measures such as efficiency and the percentage of selection reduced. Hence, more measures, such as code coverage and cost (time) incurred, can be incorporated to ascertain quality improvement. Our future work is to focus on the said issues to improve the selection quality to 10 percent.

**Supplementary Materials:** Electronic supplementary material is available online at: https://github.com/NARAYNAN888/fuzzy_program/blob/main/final_fuzzy_dataset.csv (accessed on 3 July 2023).

**Author Contributions:** S.K.R.—Original draft writing; S.G., Conceptualization; S.K.T., Data curation; A.S., Formal analysis; D.S.K., Investigation; H.K.A., Methodology; T.J.A., Validation. All authors have read and agreed to the published version of the manuscript.

**Funding:** Princess Nourah bint Abdulrahman University Researchers Supporting Project number (PNURSP2023R 384), Princess Nourah bint Abdulrahman University, Riyadh, Saudi Arabia.

**Institutional Review Board Statement:** Not Applicable.

**Data Availability Statement:** Dataset: https://github.com/NARAYNAN888/fuzzy_program/blob/main/final_fuzzy_dataset.csv (accessed on 3 July 2023) ## Code/Software, https://github.com/NARAYNAN888/fuzzy_program/commit/d0a8c94d644f881dff2636d80eb81d0135816c07 (accessed on 3 July 2023).

**Acknowledgments:** We want to acknowledge Princess Nourah bint Abdulrahman University Researchers Supporting Project's support under project number PNURSP2023R 384. This project is affiliated with Princess Nourah bint Abdulrahman University located in Riyadh, Saudi Arabia.

**Conflicts of Interest:** The authors declare no conflict of interest.

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
