# Peer review of "Test Case Selection through Novel Methodologies for Software Application Developments"

_symmetry, doi:10.3390/sym15101959_

Round 1

Reviewer 1 Report (New Reviewer)

The article is devoted to solving the problem of creating test cases. The topic of the article is relevant. The structure of the article does not correspond to that accepted in MDPI for research articles (Introduction (including analysis of analogues), Models and methods, Results, Discussion, Conclusions). The article is formatted carelessly. The level of English is acceptable. The article is easy to read. The figures in the article are of acceptable quality. The article cites 39 sources, most of which are not relevant.

The following comments and recommendations can be formulated regarding the material of the article:

1. A test case is a professional documentation of a tester, a sequence of actions aimed at testing some functionality, describing how to arrive at the actual result. I didn't see this in the article.

2. Any test case must include: - A unique test case identifier - necessary for convenient organization of storage and navigation through our test kits. - The title is the main theme or idea of the test case. Brief description of its essence. – Preconditions, i.e. description of conditions that are not directly related to the functionality being tested, but must be met. - For example, only a registered user can leave a comment on your portal. This means that for the test case “Creating a comment” it will be necessary to fulfill the preconditions “user is registered” and “user is authorized”. - Steps - a description of the sequence of actions that should lead us to the expected result. - Expected outcome - outcome: what we expect to see after completing the steps. I didn't see this in the article.

3. What should not be in a test case: 1. Dependencies on other test cases; 2. Unclear formulation of steps or expected results; 3. Lack of information necessary to pass the test case; 4. Excessive detail. Please prove that this is not the case.

4. The test case must contain all the information necessary to pass it. For example, if we check the login window on the site, then we will need a login and password, otherwise passing this script will be impossible. This is not in the article.

5. I didn’t understand where the scientific novelty is in this article. However, the authors did not position their article in any way in the classification of types of MDPI publications.

6. Refrenses contains links to articles where neural networks are used for testing. This is not in the article. Why then these links?

7. The author must clearly justify the specific features of the “software application”. Now the article claims to be “instructions for testing everything,” but this is not true either in terms of level or content.

Author Response

Reviewer 2 Report (New Reviewer)

In this manuscript, the authors proposed two classes of approaches, which can be used to help the test users to select the suitable test cases and then quicken the test process or enhance the quality of the software test. 

Basically, the idea could be interesting, and the research desing is also satisfactory. However, before the final acceptance, the following concern can be addressed,

1) The abstract is a little tedious and can be refined. In addition, the tense is a little messy, and it can be unified;

2) Some typos exist. As an example, before listing two contribution points, "Our contributions to the research work"--> "Our contributions in this work can be summaried as the following two aspects,"; in Line 423, "22.. 2", there is a redundant point.

3) All tables can be redrawn in three-line style, which will be more elegant. In addition, Table 1 is a little wider, and I suggest that the authors adopt the smaller font for the texts inside each cell.

4) In Table 7/8/9, what does mean the black box at the right-bottom coner?

In this manuscript, the authors proposed two classes of approaches, which can be used to help the test users to select the suitable test cases and then quicken the test process or enhance the quality of the software test. 

Basically, the idea could be interesting, and the research desing is also satisfactory. However, before the final acceptance, the following concern can be addressed,

1) The abstract is a little tedious and can be refined. In addition, the tense is a little messy, and it can be unified;

2) Some typos exist. As an example, before listing two contribution points, "Our contributions to the research work"--> "Our contributions in this work can be summaried as the following two aspects,"; in Line 423, "22.. 2", there is a redundant point.

3) All tables can be redrawn in three-line style, which will be more elegant. In addition, Table 1 is a little wider, and I suggest that the authors adopt the smaller font for the texts inside each cell.

4) In Table 7/8/9, what does mean the black box at the right-bottom coner?

Author Response

Reviewer 3 Report (New Reviewer)

Dear Authors,

I hope this message finds you in good health. I have thoroughly reviewed your manuscript. Overall, your research shows promise, but I have identified several areas where revisions are needed to improve clarity, structure, and overall quality.

Sincerely,

Reviewer

Here are some specific suggestions:

1. Our contributions to the research work (In page 3, line 71).

Here are my suggestions for modifying your three research contributions:

(1). Hybrid Grey-Box Testing Approach:

Title: "Introducing a Hybrid Grey-Box Testing Approach"

In the manuscript, provide a detailed explanation of how your approach combines white-box and black-box techniques.

Highlight the unique aspects of your approach that enhance testing stability, backed by quantifiable metrics such as improved test coverage or reduced regression issues.

(2). Feature-Based Test Case Selection Methodology:

Title: "Proposing a Feature-Based Test Case Selection Methodology"

Within the paper, explain how your methodology identifies critical software aspects using features marked exclusively by skilled testers.

Emphasize its versatility across various testing approaches and its advantages over existing methods, particularly in terms of quantifiable testing stability metrics.

(3). Quantum-Based Techniques for Test Case Selection:

Title: "Innovative Quantum-Based Test Case Selection Techniques"

In your manuscript, elucidate how these techniques, rooted in quantum principles, enhance testing effectiveness, especially in regression testing.

Highlight the measurable improvements in stability facilitated by these techniques.

These modifications will help you effectively convey your research contributions and underscore their distinctions from existing methods, with an emphasis on quantifiable metrics related to testing stability.

2. In Table 1, present a concise summary of the various test case selection methods mentioned in the literature review. Use brief descriptions and key points to capture the essence of each method. Include a column indicating the strengths and weaknesses of each method.

3. After presenting Table 1, provide a brief summary paragraph that highlights the key findings and trends in the literature regarding test case selection methods. Mention the common strengths observed across these methods, such as improved coverage or reduced test suite size. Highlight the limitations or weaknesses that have been identified in the literature.

4. in section 2.2 (page 9, lin3 228-230). “Our method reduces time, cost, and complexity so that the product is delivered to the customer or customers on time with all the required tests without compromise.”

The authors should offer a description of how their proposed method was practically applied in software testing scenarios. This could include case studies or experiments where the method was utilized in real software projects. Specifically, in the Results and Discussion section, the authors should present concrete data from their testing experiments to demonstrate the realized time and financial cost savings resulting from the application of their method. This data should be used to substantiate their claims regarding the effectiveness of their approach in reducing time and financial costs in software testing.

5. In section 3. Motivational Example.

(1). Conversion to a 5-Scale in Table 2: The authors should explain why a 5-scale (Very Low to Very High) was chosen for linguistic scores and why other scales were not considered. Elaborate on the advantages or specific reasons for using this scale.

(2). Selection of the 11 Features: Clarify how the 11 features listed in Table 2 were determined for inclusion in the study. Were other features considered and why were these specific features chosen? Discuss whether the inclusion of additional features was considered and whether their presence would yield different research results or insights. By providing these explanations, the authors can enhance the transparency and clarity of their research methodology and decisions.

6. In page 11, line 303. The authors should provide additional details on how Table 4 was formed to offer more insight into the research methodology and data processing.

7. In section 5. Results and Discussion.

The authors should perform a comparative analysis between their proposed method and other existing methods, especially those mentioned in the literature review. This analysis is crucial for demonstrating the effectiveness of their proposed approach. By comparing the results and performance metrics of their method with those of established techniques, the authors can provide empirical evidence of the superiority or uniqueness of their approach in improving error detection and optimizing the testing process. This comparative analysis will enhance the credibility and value of their research findings.

8. In section 6 to 7. To enhance the paper's structure and impact, the following recommendations are suggested:

(1). Merge Future Work into Conclusion: Combine the "Future Work" section with the "Conclusion" to provide a more cohesive ending. Discussing future research directions within the conclusion will offer a clearer perspective on potential extensions.

(2). Elaborate on Research Limitations: In the "Conclusion," explicitly address the research limitations to provide a balanced view of the work's scope and constraints.

Incorporating these recommendations will strengthen the paper's overall structure and effectiveness in communicating its findings to readers. The research has the potential to significantly contribute to software testing, and these adjustments will ensure the findings are well-conveyed.

Round 2

Reviewer 1 Report (New Reviewer)

I formulated the following comments to the previous version of the article:

1. A test case is a professional documentation of a tester, a sequence of actions aimed at testing some functionality, describing how to arrive at the actual result. I didn't see this in the article.

2. Any test case must include: - A unique test case identifier - necessary for convenient organization of storage and navigation through our test kits. - The title is the main theme or idea of the test case. Brief description of its essence. – Preconditions, i.e. description of conditions that are not directly related to the functionality being tested, but must be met. - For example, only a registered user can leave a comment on your portal. This means that for the test case “Creating a comment” it will be necessary to fulfill the preconditions “user is registered” and “user is authorized”. - Steps - a description of the sequence of actions that should lead us to the expected result. - Expected outcome - outcome: what we expect to see after completing the steps. I didn't see this in the article.

3. What should not be in a test case: 1. Dependencies on other test cases; 2. Unclear formulation of steps or expected results; 3. Lack of information necessary to pass the test case; 4. Excessive detail. Please prove that this is not the case.

4. The test case must contain all the information necessary to pass it. For example, if we check the login window on the site, then we will need a login and password, otherwise passing this script will be impossible. This is not in the article.

5. I didn’t understand where the scientific novelty is in this article. However, the authors did not position their article in any way in the classification of types of MDPI publications.

6. Refrenses contains links to articles where neural networks are used for testing. This is not in the article. Why then these links?

7. The author must clearly justify the specific features of the “software application”. Now the article claims to be “instructions for testing everything,” but this is not true either in terms of level or content.The authors responded to all my comments. I found their answers quite convincing. I support the publication of the current version of the article. I wish the authors creative success.

Author Response

Reviewer 2 Report (New Reviewer)

The authors have made the necessary revisions, and the current edition can be accepted at present.

Author Response

Reviewer 3 Report (New Reviewer)

Dear Authors,

I have reviewed the revised version of the manuscript titled "Test case selection through novel methodologies for software application developments" (Manuscript ID: symmetry-2628385). The manuscript has made significant improvements based on my previous suggestions, addressing major issues. However, I have identified some minor issues, including inconsistent table formatting and incorrect table ordering.

I recommend accepting the manuscript once these minor revisions are made.

Best regards,

Reviewer

Here are my comments.

1. Page 29, line 853-859. The reference should be corrected to Table 15.

2. The tables in the manuscript should consistently adhere to the format of having only three horizontal lines.

Minor editing of English language required.

Author Response

This manuscript is a resubmission of an earlier submission. The following is a list of the peer review reports and author responses from that submission.

Round 1

Reviewer 1 Report

The comments as follows :

This manuscript does an excellent job of demonstrating significance. This work is a good reminder for all. Please follow the science instruction format, especially in references.

Abstract: Please focus the abstract on your study and your results.
The authors should specify more details regarding the Experiment for the proposed algorithm. The authors should provide more details regarding the analysis of the results.

How to initialize the agents in the proposed Algorithm?

Some additional experiments are required:
  - Runtime

It is necessary to discuss the complexity of the proposed Algorithm. Read and cite these references.
M. Hassib, I. El-Desouky, M. Labib and E. -S .M. El-kenawy, “WOA + BRNN: An imbalanced big data classification framework using Whale optimization and deep neural network,” Soft Computing, vol. 24, no. 1, pp. 5573–5592, 2020. DOI 10.1109/ACCESS.2019.2955983

Some syntax errors or improper expressions exist in the manuscript. More up-to-date studies are suggested to be cited.

Reviewer 2 Report

See the attached file. Thanks.

Reviewer 3 Report

Authors

The “Abstract” is quite long. It should be shortened, as there is a lot of information that is unnecessary in the abstract.

The terminology of the proposed model, although explained adequately, has an abbreviation that is difficult to memorize: DSFRCRTSCFS.

The introduction is completely done without the support of a single reference. In scientific work, what has already been done by researchers in the past must always be referred to. Therefore, the introduction is not sound. At least in the approach, the themes must be reinforced with published literature.

At the structural level, subsection 1.1 is orphaned and comes inappropriately. A subsection should be created right after the introduction giving the framework of the theme.

Section 2 should be replaced by “literature review”.

The way in which literature is introduced into the text is inadequate. Authors should review other published articles and follow an identical standard.

The literature presented is quite scarce. Namely, many of fuzzy analysis is referred, but only one reference of applied literature is presented. In addition to table 1, which brings together the entire literature of the manuscript (only 26 article references and only articles 4 and 7 referring to fuzzy sets). I suggest that more works developed internationally on this subject, in different areas, be referred to reinforce the diversity of the application of this type of approach. Suggested “applied fuzzy set” literature to add:

- Mocq, J., St-Hilaire, A., & Cunjak, R. A. (2015). Influences of experts' personal experiences in fuzzy logic modeling of Atlantic salmon habitat. North American Journal of Fisheries Management, 35(2), 271-280.

- Ramos, J., Lino, P. G., Caetano, M., Pereira, F., Gaspar, M., & dos Santos, M. N. (2015). Perceived impact of offshore aquaculture area on small-scale fisheries: A fuzzy logic model approach. Fisheries Research, 170, 217-227.

- Vásquez, R.P., Aguilar-Lasserre, A.A., López-Segura, M.V., Rivero, L.C., Rodríguez-Duran, A.A., & Rojas-Luna, M.A. (2019). Expert system based on a fuzzy logic model for the analysis of the sustainable livestock production dynamic system. Computers and Electronics in Agriculture, 161, 104-120.

Section 5 “results and discussions” sounds strange. “Discussion” is what makes sense (in the singular).

Reviewer 4 Report

The title of the article is relevant enough. Potentially, the problem of time optimization in testing needs to be solved.

DSFRCRTSCFS is a very long abbreviation for the name of the model.

There is an inaccuracy in the introduction, because after the developer corrects the errors found earlier during testing, it is necessary to re-test the corrected pieces of the program code, and not selectively or at random.

“Yet another approach adopted by practitioners and researchers is generating test cases with respect to functional and non-functional requirements” does not apply to code testing. It's about requirements verification. Testing and verification are different approaches. The author should clearly limit the scope of the problems that he solves. This is very important for the article.

The material of the article from 2. Related Work to 2.1 Analysis of a Literature Work to Find the Gap in Research looks not quite finished. This part lists only the methods without specifying their weaknesses or strengths, and then appears Table 1. Literature survey in the domain of test case selection, which provides the same list of methods. There should be a result of the analysis of methods, but it is not in the article. It is also worth in table 1 to maintain the chronology of the appearance of methods by year. For example, start listing from earlier years, and end with later ones.

At the beginning of 2.1 Analysis of a Literature Work to Find the Gap in Research, the methods are listed again. Why are they listed? Where is their analysis again? The enumeration ends with the question Ask, do these methods really reduce complexity and time? Such questions are not good style for writing a scientific article.

Further, after the question, the authors add continue "On the other hand". And where is the beginning, i.e. On the one side?

This part of "The main question here is, did all the factors help the software developer to reduce cost, time and complexity?" From what I can see, I can safely say there is some value in these methods and their results.” It looks very strange. The authors ask themselves a question and, without any arguments or results of analysis, answer themselves. This is not 100% scientific style.

The authors want to optimize i.e. reduce testing time. There are no arguments to suggest even that such a problem exists. The existing methods are not good enough, at least in terms of time costs. Reading the article, one might even think that there are no problems in testing methods in terms of time.

I.e. there is no analysis of existing methods in the article.

In section 3. Motivational Example Table 2 shows 11 different features (С1-С11): C1-Critical bugs, C2-Requirements covered , C3-Time, C4-LOC covered, C5-Bugs detected,C6-Length, C7-Critical requirements covered, C8-Customer priority, C9-Fault proneness C10-Requirements volatility and C11-Implementation complexity. What are these features? There is no description for them. Why exactly 11 and not 10 or 15 or 20? There is no rationale for choosing these features.

The application of these methods has not been explicitly demonstrated. The article lacks graphic elements (pictures) that would demonstrate the integration of methods.

The English language requires substantial and stylistic revision.

Round 2

Reviewer 2 Report

See the attached file. Thanks.

English very difficult to understand/incomprehensible.

Reviewer 3 Report

Round 2 Authors

Most of the suggestions were fulfilled. The manuscript benefited immensely from this, particularly in those parts where there was a complete lack of literature consulted.

There are, however, still some small inadequacies that need to be improved and that are still too recurrent. Please check this and correct it, otherwise the manuscript will not be appealing, and consequently the chances of being published will decrease.

For example, references in the main body of the text must include the name of the first author followed by “et al.”

For example, on lines 97-8 there are two problems: It says “Siavash Mirarab, Soroush Akhlaghi, and Ladan Tahvildari [9]” it should say “Mirarab et al. [10]”. 1st problem - The first and last names of the authors should not be included in the text, it should only the surname of the first author followed by “et al.”; 2º The reference is not to [9] but to [10].

There are several other cases throughout the text that should be reviewed and corrected according to the same procedure.

From what was seen above, cited literature should be checked if the references are numbered accordingly, i.e., if the number that appears in the text corresponds to the one that appears in the final section of the literature (“References” section).

Throughout the body of the text there are sections that are too long in the same paragraph. Authors should consider making paragraphs shorter and dividing them according to the subtopics addressed in those sections. Better editing of the text should also be done, as the part where punctuation is not given much attention, which makes readability difficult and does not make the manuscript attractive.

The new section 2 “Literature review” is a huge monobloc. It shouldn't be like that. It should be separated into several paragraphs, according to the topics consulted. Furthermore, it should come with 2 subsections and not with the orphaned subsection “2.1 Analysis of a Literature Work to Find the Gap in Research”. There should be a first subsection just below “2. Literature Review” one that is something like e.g. “2.1 Mainstream literature in the field“ and then yes but with a different numbering “2.2 Analysis of a Literature Work to Find the Gap in Research”.

Reviewer 4 Report

Thank you for considering my previous recommendations, but not all. The style of presentation should be more scientific of course. This is still worth working on.

The level of scientific English can be further improved.
